# Optimal number of charging station and pricing strategy for the electric vehicle with component commonality considering consumer range anxiety

**Wenchao Yu[1], Linghong Zhang[1,2]\*, Rui Lu[3], Junjie Ma[4]**

**1** Business School, Shandong Normal University, Ji'nan, China, **2** Management Science and Engineering Postdoctoral Mobile Station, Shandong Normal University, Ji'nan, China, **3** School of Economics and Management, Hangzhou Normal University, Hangzhou, China, **4** Law School, Tongji University, Shanghai, China

\* 614104@sdnu.edu.cn

**Data Availability Statement:** All relevant data are within the paper and its Supporting information files.

## Abstract

The battery driving mileage on a single charge and convenience of the charging stations affect Electric Vehicle's (EV) demand. This paper studies the optimal number of charging stations and EV's price strategy considering different component commonality configurations. Assume the EV manufacturer provides two types of EV and the two EVs have the same battery configuration (battery as a common part) or the same naked vehicle–EV without batteries (naked vehicle as a common part). And the common part could be configured with low or high quality. We discuss four scenarios with different common parts and different quality levels. For each scenario, we present the optimal number of the charging stations and EV prices. Then we compare the optimal solutions and manufacturer's profits in above four scenarios with numerical simulation and give some managerial insights. Our analysis reveals that (1) consumers' range anxiety towards battery will affect manufacturer's product configuration strategy, EVs' prices and demands. (2) large consumers' sensitivity towards charging station will corresponding to more charging station, high EV prices and demands. If consumers are very concerned about the charging convenience, high-end electric vehicles need to be launched first, then as customers' anxiety about charging decreases, the low quality EV could be developed and diffused. (3) the unit product cost reduction caused by the commonality may increase or decrease the EVs' prices, which depends on the relationship between the demand increment incurred by one more charging station and the cost coefficient of building the charging station. (4) The low quality naked vehicle as common component will increase both the number of the charging stations and the demand and the manufacturer is more likely to obtain high profits. (5) the cost saving coefficient of battery common parts has greater influence on the selection of commonality. When consumers' range anxiety towards battery is very high, manufacturers should choose low-quality naked vehicles or high-quality battery as common components.

**Funding:** This work was supported by the National Natural Science Foundation of China [grant numbers 71602103 and 41971252]. The funders provided support for the cost of enterprises' survey in the process of study design, and provided support for the publication of papers.

**Competing interests:** The authors have declared that no competing interests exist.

## Introduction

More and more consumers are beginning to accept electric vehicles, and the sales of electric vehicles have been increasing in recent years. The global electric vehicles sales, including the pure electric vehicles and the plug-in hybrid vehicle, has soared from 2.046 million to 10.65 million from 2019 to 2022 (Chen, 2023 [1]). However, compared with global car sales totaled 81 million in 2022, the EV manufacturer still need make more efforts to let most consumers purchase electric vehicles.

There are two main factors that influence consumer choice: EV price and the range anxiety. The EV price is higher than the gasoline vehicle at least 20% (Wu et al., 2019 [2]). Take Volkswagen's Golf as an example, the pure electric golf is about 22.5 thousand dollars and the gasoline golf with same model is 18.2 thousand dollars. The range anxiety is another obstacle that affects consumer purchasing EVs (Guo et al., 2018 [3], Tanţău & Gavrilescu, 2019 [4], Xu et al., 2020 [5]). Limited battery range and insufficient charging infrastructure can cause inconvenience and range anxiety (Davidov, 2020 [6]).

The EVs' price is mainly related to the naked vehicle and battery configuration. Using the common components for multiple models is a good way to reduce the vehicles' production costs even though the commonality's R&D still needs invest some money(Heese and Swaminathan, 2006 [7]). For example, in 2021 NIO launched ET5 and ET7 in China, ET5 and ET7 are configured with the same long life battery (the longest range is 1000 kilometers per charge) and different naked vehicle. Tesla produced Model Y that has two configurations–the standard battery life version and the long range version with the same naked vehicle. Note that the NIO and Tesla chose different components for the two products as the commonality–NIO chose the battery, Tesla chose the naked vehicle. Motivated by the two manufacturers' commonality configuration, we try to explore how should EV manufacturers choose common parts, and the impact of the selection of common parts on product prices, sales and corporate profits.

Range anxiety is mainly divided into limited battery range and inconvenient charging station. At present, many EV manufacturers have increased their investment in battery research and development to increase battery range (Ke and Chen, 2022 [8], Chen and Fan, 2020 [9]). For example, BYD invested 1.5 billion dollars to build a 40GWh new battery production line. In addition, some manufacturers, such as Tesla, have begun to establish their own brand charging stations to improve the charging convenience, in order to increase EVS sales. Both investing in battery R&D and constructing the charging stations will increase the cost and price of electric vehicles. By balancing the costs of battery R&D and the charging station's construction and the revenue from increased sales, how the EV manufacturer should configure the battery and whether and how many charging stations should be constructed are the significant problems worth studying. Based on above EV manufacturer's commonality selection, the battery R&D and charging station construction reality, we mainly focus on following questions:

(1) Which components (battery or naked vehicle) should be used as a commonality when the EV manufacturers provides two types of EVs? And how should the manufacturer set the commonality quality level, i.e., high or low quality?

(2) Whether the manufacturer should construct charging station? If yes, how many charging stations should be constructed?

(3) How the component commonality's quality and the cost coefficient of the charging stations affect EV prices?

In order to address above questions, we assume that the EV manufacturer provides two types of EVs and discuss two basic scenarios: using battery and naked vehicle as common parts for the two types of EVs, respectively. And in each scenario, we consider the common part's quality could be high or low. For each scenario, we present the optimal number of the charging stations and EVs' prices, then we compare the optimal solutions and discuss effects of commonality on manufacturer's decisions in different scenarios. Further, we present the EV manufacturer's optimal commonality strategy by numerical simulations. Finally, we give the managerial insights.

Our paper mainly has three contributions. First, we divide the range anxiety into battery range anxiety and charging convenience anxiety and both of them affect the EV demand, while most other studies only consider the one type of anxiety (Davidov, 2020 [6], Guo et al., 2018 [3], Luo et al., 2020 [10], Tanţău and Gavrilescu, 2019 [4], Xu et al., 2020 [5], Zhang et al., 2021 [11]). Analyzing the range anxiety in detail will allow manufacturers to take more targeted measures to boost EV demand and make more profits. Second, we incorporate the component commonality into EV manufacturer's decision process, while most other studies focus on how to choose battery quality or how to set the number of the charging stations (Bi et al., 2019 [12], Bonges and Lusk, 2016 [13], Csiszár, 2019 [14], Zhu et al., 2018 [15]). The commonality design is an important decision for automobiles, especially considering the high cost of developing a production line (about 20 million per product line for vehicles). Third, we discuss the mutual effect between improving battery range and increasing the number of the charging stations. Understanding the relationship between the two measures will avoid the EV manufacturer spend extra money on easing range anxiety and make more profit.

The structure of this paper is as follows: In Section 2, we give the literature review. In Section 3, we present the model formulation. In Section 4, we present the four scenarios with different commonalities and provide the optimal solutions. Then we compare the different scenarios and give the suggestions for the manufacturer. In Section 5, we conclude our research and give the future research orientations.

## Literature review

Our work is related to the following two streams of literature. The first stream is the development of electric vehicles and the other is the impact of product commonality on the product design.

### Electric vehicle

Literature on electric vehicles mainly concentrated on the pricing problems, construction of charging stations and battery range. Firstly, we review studies concerning pricing problems. He et al. (2017) [16] analyzed the critical price of automobile from the perspective of whole life cycle. Liu et al. (2019) [17] indicated that electric vehicles will have an advantage over fuel vehicles when battery prices fall. Ke & Chen (2022) [8] found consumer trust positively affects the demand of the electric vehicles using the manufacturer-made batteries and then affects the manufacturer's battery R&D decision. Yang et al. (2019) [18] proposed a pricing model of EV sharing subsidy. Zhang et al. (2020) [19] studied the pricing strategy for electric vehicle dynamic charging. Allahmoradi et al. (2022) [20] showed that reducing the price of electric vehicles and fuel vehicles can effectively stimulate the market demand for electric vehicles. Guo et al. (2023) [21] analyzed that compared with fuel vehicles, electric vehicles have no cost advantage due to the high purchase cost. Kondev et al. (2023) [22] showed that when the vehicle is more powerful, electric vehicles have a price advantage over internal combustion engine vehicles.

Then, we review studies concerning construction of charging stations. Zhang et al. (2019) [23] explored the optimal product choice for the manufacturer. Heydari et al. (2020) [24] studied pricing and green quality issues considering consumer environmental awareness. Mak et al. (2013) [25] employed a robust optimization framework to deploy the battery swapping infrastructure. Kim et al. (2021) [26] examined manufacturers' trade-off between green car prices and technological improvements. Niu et al. (2021) [27] researched on automotive supply chain structure. Sun et al. (2019) [28] introduced a multi-interval battery charging station location-inventory and recharging planning problem for electric vehicles (BSS-LIRP).

Another important issue related to electric vehicles is the battery range. Seiho et al. (2017) [29] showed that mileage is an important factor affecting the market share of electric vehicles. Teichert et al. (2019) [30] proposed a method to optimize the design of battery packs under specific operating conditions. Juul (2012) [31] studied on battery capacity and price sensitivity related issues in power systems. Junquera et al. (2016) [32] explored the key factors influencing consumers' purchase of electric vehicles. This study can not only provide reference for consumers' choice and purchase, but also provide theoretical basis for the popularity of electric vehicles. Fan et al. (2020) [33] analyzed an electric vehicle manufacturer's product choice strategy of producing an electric vehicle with a low driving range only, with a high driving range only, and with both driving ranges. The results show that consumers' mileage anxiety, battery production costs and other factors will affect the manufacturer's product selection decision.

While there are many studies on the related issues of charging station operation, there is still a lack of comprehensive studies on charging stations considering common components. We consider the battery as a candidate for common components. Unlike other components, the common battery will affect the use of related infrastructure. In this context, it is necessary to study the optimal number of charging stations. Considering consumers' range anxiety, we study the choice of battery quality.

## Product commonality

The competitiveness in today's market forces many companies to rethink the way they design products. Instead of developing one product at a time, many manufacturing companies are developing families of products to provide enough variety for the market while keeping costs relatively low. For example, Citroen's single-box multi-function sedan Picasso and sedan Sara, although they are very different in appearance, belong to the same platform. Thevenot et al. (2007) [34] proposed a design method to effectively balance the commonality and diversity of product families. Blecker & Abdelkafi (2007) [35] introduced a generality index for evaluating the generality of product families. Ciravegna et al. (2013) [36] conducted research on the relative advantages and risks of corporate outsourcing. Song & Zhao (2009) [37] found that although component commonality is in general beneficial, its value depends strongly on component costs, lead times, and dynamic allocation rules. Under certain conditions, several previous findings based on static models do not hold. In particular, component commonality does not always generate inventory benefits under certain commonly used allocation rules. Mohebbi & Choobineh (2005) [38] found that component commonality significantly interacts with existence of demand and supply chain uncertainties, and benefits of component commonality are most pronounced when both uncertainties exist. Bernstein et al. (2011) [39] investigated the effect of commonality on product variety and compare its benefits under different demand characteristics.

Component commonality can help companies reduce the cost of providing product variety to their customers. However, determining the extent to which component commonality should be used is difficult. Thonemann & Brandeau (2000) [40] present an approach to

determine the optimal level of component commonality for end-product components that do not differentiate models from the customer's perspective. Desai et al. (2001) [41] analyzed different situations of common parts from the perspective of cost and revenue. Dangayach & Deshmukh (2001) [42] considered the manufacturing and improvement activities and other factors to analyze the strategic manufacturing activities of automobile enterprises. Takai & Sengupta (2017) [43] took the engine of electric bicycle as an example to evaluate the profitability evaluation method of common parts.

In addition, commonality makes the distinction between products aimed at different consumer groups smaller and makes consumer switching between products more likely such that cannibalization is always intensified. The literature on common strategies that exacerbate cannibalism is mostly based on a key assumption that for all product attributes, the quality estimate of one consumer segment is greater than the quality estimate of another segment; that is, the preference structure of a segment dominates the preference structure of another segment. Kim et al. (2013) [44] considered a non-dominant preference structure, in which each segment has a more valuable attribute and the other part does not. Wong et al. (2019) [45] investigated a manufacturer's optimal decisions in relation to the adoption of the commonality strategy in a decentralized channel as opposed to a centralized channel and find that commonality always help reduce the extent of quality distortion encountered by the low-valuation segment, regardless of the channel structure. Zhang et al. (2015) [46] investigated the condition under which component commonality is a profitable product design strategy for a firm by considering customer-choice behavior in the supply chain environment.

However, few studies focused on the automobile industry and discussed the effects of the common part's quality on price and the number of the charging station considering the consumers' range anxiety. Therefore, in this paper we combine the common components and range anxiety to discuss the optimal EVs' prices and the number of the charging station when the manufacturer chooses high/low naked vehicle or high/low battery as common components. We summarize the difference between this study and the most relevant papers in Table 1.

## Model formulation

A monopoly manufacturer sells two types of EVs to a market. We assume that the EV consists of two parts: the naked vehicle and the battery. The manufacturer may choose either naked vehicle or battery as commonality, and the commonality could be high-quality or low-quality. For example, two EVs could be configured with same battery, and the battery could be standard range battery (low quality) or long range battery (high quality). We denote the naked vehicle and battery quality as $q_1^{j_1}$ and $q_2^{j_2}$, respectively, where $j_1, j_2 = H$ or $L$ represents high quality or low quality and $0 < q_1^{j_n}, q_2^{j_n} < 1$, $n = 1$ or 2. We divide the market into two segments: the

**Table 1. Differences between this research and the most relevant papers.**

| Articles | EV | Battery Quality | Bare vehicle quality | Mileage anxiety | Convenience of swap station | Product commonality |
|---|---|---|---|---|---|---|
| Bonges & Lusk [13] | √ | | | √ | √ | |
| Chen & Fan [9] | √ | | | √ | | |
| Zhang & Huang [47] | √ | | | √ | | |
| Lu et al. [48] | √ | | | | | √ |
| Fan et al. [33] | √ | √ | | √ | | |
| Kuppusamy et al. [49] | √ | | | √ | √ | |
| This paper | √ | √ | √ | √ | √ | √ |

high-end market and the low-end market, and the potential demands of each segment are denoted as $a_1$ and $a_2$. The demand in each segment is affected by EV price, $p$, battery quality and the number of the charging station, $r$. The demand function of each segment is as follows:

$$D_j \;=\; a_j - p_j - \theta(1 - q_2^{j_2}) + \delta r \quad where \; j = 1 \; or \; 2 \tag{1}$$

Considering that the range anxiety towards battery will go away only when the battery has unlimited range per charge, that is, $q_2^{j_2} = 1$, hence, the term $1 - q_2^{j_2}$ could be explained as the inconvenience caused by battery limited range per charge in the demand function and $\theta$ represents consumer's mileage anxiety caused by battery. Because both the two EVs could use the charging station built by the manufacturer, thus we assume the number of the charging station affects two EVs' demand simultaneously. $\delta$ represents the consumers' sensitivity coefficient towards the number of charging stations. The EV demand decreases with the price and increases with the battery maximum range per charge and the number of charging stations.

Consistent with Desai et al. (2001) [41], we assume that the cost is a convex function of the naked vehicle quality, $q_1^{j_1}$, and battery quality, $q_2^{j_2}$. For mathematical tractability, we assume a quadratic function, so the per unit production cost of component $i$ with quality $q^j$ is $c_i q_i^{j^2}$, where $c_i > 0$ is a cost coefficient that reflects the differences in cost of producing quality across different component types (Heese and Swaminathan, 2006, [7]), $i = 1, 2$ represents the EV naked vehicle, battery.

Similar to Desai et al. (2001) [41], assume that with the commonality design effort $e_i$, the unit production cost of a component of quality $q_i^{j_2}$ is $c_i q_i^{j_2 2} - c^i \sqrt{e_i}$. Then the EV's cost function with commonality i and its quality levels $(j_1, j_2)$ is

$$C_i^{j_1 j_2} \;=\; c_1 q_1^{j_1 2} + c_2 q_2^{j_2 2} - c^i \sqrt{e_i} \;=\; c^{j_1 j_2} - c\sqrt{e_i} \tag{2}$$

where $c$ is the saving cost coefficient per unit product with commonality.

We summarize the main parameters and decision variables that will be used in the the model in Table 2.

Note: $i = 1, 2, 3, 4$ represent high-quality common naked vehicle, low-quality common naked vehicle, high-quality common battery, low-quality common battery, respectively.

**Table 2. Model parameters and decision variables.**

| Parameter | Explanation |
|---|---|
| $a$ | market potential |
| $\theta$ | consumer's range anxiety towards battery |
| $\delta$ | consumers' sensitivity coefficient towards the number of charging stations |
| $k$ | cost coefficient for the construction of charging station |
| $\pi_j$ | manufacturer's profits in scenario $j$ |
| $q_1^{j_1}, q_2^{j_2}$ | qualities of the naked vehicle and battery |
| $D_1, D_2$ | demand functions of product 1 and 2 |
| $C_1, C_2$ | cost functions of product 1 and 2 |
| $e_1, e_2$ | design efforts to unify naked vehicle and battery |
| $c, c'$ | cost reduction coefficients per unit product with commonality |
| $c_1, c_2$ | cost coefficients of naked vehicle and battery |
| **Decision variable** | **Explanation** |
| $p_1$ | Product 1's price |
| $p_2$ | Product 2's price |
| $r$ | the number of charging stations |

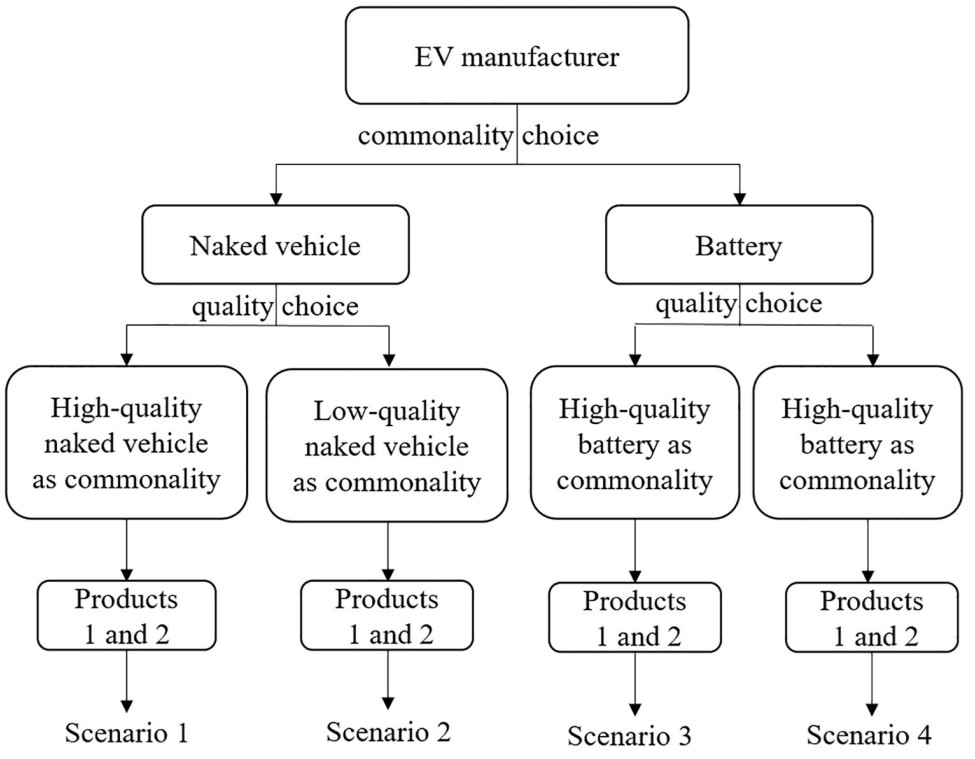

**Fig 1. Supply chain structure under different scenarios.**

## Results

In this section, we first give the optimal solutions when the manufacturer makes the naked vehicle as common component, then we discuss the scenario that the battery as a commonality. Further we analyze the influences of parameters on optimal decisions and compare the optimal strategy in different scenarios. Fig 1 shows supply chain structure under different scenarios.

### Naked vehicles as common components

In this subsection, we consider that the manufacturer chooses the naked vehicle as a commonality, and the common naked vehicle could be high quality or low quality.

**Scenario 1: High-quality naked vehicles as common components.** When the EV manufacturer provides two types of EVs to the market and the two products has the same naked vehicle with high quality level, then the demand functions of the product with high quality vehicle and battery and the product with high quality vehicle and low quality battery are as follows:

$$D_1 = a_1 - p_1 - \theta(1 - q_2^H) + \delta r \tag{3}$$

$$D_2 = a_2 - p_2 - \theta(1 - q_2^L) + \delta r \tag{4}$$

The manufacturer's profit is:

$$\pi_1 = (p_1 - C_1)D_1 + (p_2 - C_2)D_2 - e_1 - \frac{1}{2}kr^2 \tag{5}$$

where,

$$C_1 = c_1 q_1^{H2} + c_2 q_2^{H2} - c\sqrt{e_1} = c^{HH} - c\sqrt{e_1} \tag{6}$$

$$C_2 = c_1 q_1^{H2} + c_2 q_2^{L2} - c\sqrt{e_1} = c^{HL} - c\sqrt{e_1} \tag{7}$$

**Theorem 1.** In scenario 1, the optimal number of charging stations, the prices of products are given as follows:

$$\dot{r} = \frac{\delta(a_1 + a_2 - c^{HH} - c^{HL} - 2\theta + q_2^H \theta + q_2^L \theta + 2c\sqrt{e_1})}{2(k - \delta^2)} \tag{8}$$

$$\dot{p}_1 = \frac{a_1 + c^{HH} - c\sqrt{e_1} + \theta(q_2^H - 1)}{2} + \frac{\delta^2(a_1 + a_2 - c^{HH} - c^{HL} + 2c\sqrt{e_1} + \theta(q_2^H + q_2^L - 2))}{4(k - \delta^2)} \tag{9}$$

$$\dot{p}_2 = \frac{a_2 + c^{HL} - c\sqrt{e_1} + \theta(q_2^L - 1)}{2} + \frac{\delta^2(a_1 + a_2 - c^{HH} - c^{HL} + 2c\sqrt{e_1} + \theta(q_2^H + q_2^L - 2))}{4(k - \delta^2)} \tag{10}$$

**Scenario 2: Low-quality naked vehicles as common components.** When the EV manufacturer provides two types of EVs to the market and make the low quality naked vehicle as a common component, then the demand functions of the product with low quality vehicle and high quality battery and the product with low quality vehicle and battery are same with the demand functions in scenario 1. Because the market demand is only affected by the battery quality and the number of the charging stations.

In this scenario, the manufacturer's profit is as follows:

$$\pi_2 = (p_1 - C_1)D_1 + (p_2 - C_2)D_2 - e_1 - \frac{1}{2}kr^2 \tag{11}$$

where,

$$C_1 = c_1 q_1^{L2} + c_2 q_2^{H2} - c\sqrt{e_1} = c^{LH} - c\sqrt{e_1} \tag{12}$$

$$C_2 = c_1 q_1^{L2} + c_2 q_2^{L2} - c\sqrt{e_1} = c^{LL} - c\sqrt{e_1} \tag{13}$$

**Theorem 2.** In scenario 2, the optimal number of charging stations, the prices of products in the high-end market and low-end market are given as follows:

$$\ddot{r} = \frac{\delta(a_1 + a_2 - c^{LH} - c^{LL} - 2\theta + q_2^H \theta + q_2^L \theta + 2c\sqrt{e_1})}{2(k - \delta^2)} \tag{14}$$

$$\ddot{p}_1 = \frac{a_1 + c^{LH} - c\sqrt{e_1} + \theta(q_2^H - 1)}{2} + \frac{\delta^2(a_1 + a_2 - c^{LH} - c^{LL} + 2c\sqrt{e_1} + \theta(q_2^H + q_2^L - 2))}{4(k - \delta^2)} \tag{15}$$

$$\ddot{p}_2 = \frac{a_2 + c^{LL} - c\sqrt{e_1} + \theta(q_2^L - 1)}{2} + \frac{\delta^2(a_1 + a_2 - c^{LH} - c^{LL} + 2c\sqrt{e_1} + \theta(q_2^H + q_2^L - 2))}{4(k - \delta^2)} \tag{16}$$

By comparing the equilibrium solutions, we can obtain the following propositions.
**Proposition 1.** In scenarios 1 and 2,

(1) $\dot{r} < \ddot{r}$ ;

(2) if $0 < \delta < \sqrt{\frac{k}{2}}$, then $\dot{p_1} > \ddot{p_1}$ , $\dot{p_2} > \ddot{p_2}$ ; if $\sqrt{\frac{k}{2}} < \delta < \sqrt{k}$, then $\dot{p_1} < \ddot{p_1}$ , $\dot{p_2} < \ddot{p_2}$ ;

(3) $\dot{D_1} < \ddot{D_1}$ , $\dot{D_2} < \ddot{D_2}$ .

Proposition 1 (1) indicates that when low-quality naked vehicles are used as common components, the optimal number of charging stations is more than the number of cases where high-quality naked vehicles are used as common components. Because when the manufacturer uses the low-quality naked vehicles as a common component, the EVs' demands are larger than those with high-quality commonality (Proposition 1(3)). Hence, the number of the charging stations when the manufacturer makes low-quality naked vehicle as common component is larger.

Interestingly, Proposition 1(2) shows when the demand increment incurred by increasing one charging station is larger than a threshold ($\sqrt{k/2}$), then the EVs with low quality naked vehicle commonality have higher prices than EVs with high quality naked vehicle commonality. Otherwise, the EVs prices are lower with low quality commonality. The main reason is that if one charging station causes an excessive increase in demand, then prices tend to rise.

**Proposition 2.** In scenarios 1 and 2,

(1) $\dot{r}$ and $\ddot{r}$ decrease with $\theta$;

(2) $\dot{p_1}$ , $\dot{p_2}$ and $\ddot{p_1}$ , $\ddot{p_2}$ decrease with $\theta$.

(3) $\dot{D_1}$ , $\dot{D_2}$ , $\ddot{D_1}$ and $\ddot{D_2}$ decrease with $\theta$.

Proposition 2 demonstrates that the number of the charging stations and EVs' prices and demands decrease with consumers' range anxiety towards battery increases. If the level of consumer anxiety towards battery is very high, then the EV manufacturer could invest more money on battery quality improvement rather than building charging station.

**Proposition 3.** In scenarios 1 and 2,

(1) $\dot{r}$ and $\ddot{r}$ increase with $c$.

(2) when $0 < \delta < \sqrt{\frac{k}{2}}$, then $\dot{p_1}$ , $\dot{p_2}$ , $\ddot{p_1}$ and $\ddot{p_2}$ decrease with $c$; and when $\sqrt{\frac{k}{2}} < \delta < \sqrt{k}$, then $\dot{p_1}$ , $\dot{p_2}$ , $\ddot{p_1}$ and $\ddot{p_2}$ increase with $c$.

(3) $\dot{D_1}$ , $\dot{D_2}$ , $\ddot{D_1}$ and $\ddot{D_2}$ increase with $c$.

Proposition 3 shows that the change trends of the optimal solutions with the unit product cost reduction due to the use of common parts. The demands increase with the decreasing of the unit EV cost, then the number of the charging station increases. For the price, similar to Proposition 1(2), the demand increment incurred by one more charging station is larger than a threshold ($\sqrt{k/2}$), then the prices increase with the unit product cost reduction. Otherwise, the prices decreases with $c$.

**Proposition 4.** In Scenarios 1 and 2,

(1) $\dot{r}$ and $\ddot{r}$ increase with $\delta$;

(2) $\dot{p_1}$ , $\dot{p_2}$ , $\ddot{p_1}$ and $\ddot{p_2}$ increase with $\delta$;

(3) $\dot{D_1}$ , $\dot{D_2}$ , $\ddot{D_1}$ and $\ddot{D_2}$ increase with $\delta$.

**Proposition 5.** In Scenarios 1 and 2,

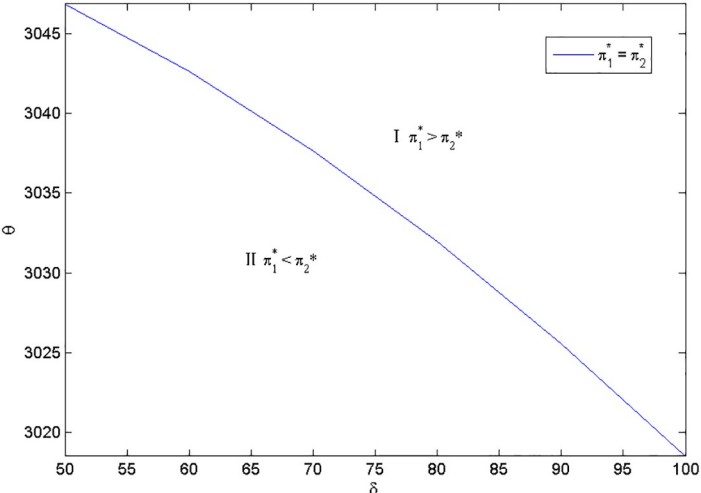

**Fig 2. Optimal naked vehicle common components selection under $c_1 = 1000$.**

(1) $\dot{r}$ and $\ddot{r}$ decrease with $k$;

(2) $\dot{p}_1$, $\dot{p}_2$, $\ddot{p}_1$ and $\ddot{p}_2$ decrease with $k$.

(3) $\dot{D}_1$, $\dot{D}_2$, $\ddot{D}_1$ and $\ddot{D}_2$ decrease with $k$.

Propositions 4 and 5 show that the number of the charging station and EVs' prices and demands increase with the consumer sensitivity towards the number of the charging station and decrease with the cost coefficient of building charging station. In other words, the more sensitive car owners are to the convenience of charging stations, the more charging stations the EV manufacturer will build, and the higher the EVs' prices. The larger cost coefficient of building charging stations, the less number of the charging station and the lower demands and prices.

Considering the complexity of manufacturer's profit functions in Scenarios 1 and 2, we use the numerical examples (Figs 2 and 3) to compare the profits in the two scenarios and present the discussion. The parameters are specified as follows: $a_1 = a_2 = 2000$, $e_1 = 800$, $c_2 = 2000$, $q_1^H = 0.8$, $q_1^L = 0.5$, $q_2^H = 0.7$, $q_2^L = 0.5$, $c = 10$, $k = 20000$.

Figs 2 and 3 shows that the manufacturer's optimal product strategy when consumers's anxiety towards battery and charging stations are different. When consumers' sensitivity towards charging station or the consumers' anxiety towards the battery quality is small, then the low quality naked vehicle as the common part will bring much more money for the manufacturer (Zone II). Otherwise, the high quality naked vehicle as the commonality is a better choice for the EV manufacturer (Zone I). Therefore, the manufacturer should decide the quality of the product according to the characteristics of the consumer. At this point, blindly pursuing high quality products does not necessarily bring good benefits. The results are in line with reality: when consumers are very concerned about the charging convenience, high-end electric vehicles need to be launched first, such as Tesla's product introduction strategy. As customers' anxiety about charging decreases, the low quality EV could be developed and diffused. Also, with different cost coefficients of naked vehicle, the overall trend in both figures is consistent. This means that small changes in the cost factor will have no effect on manufacturers' optimal choice.

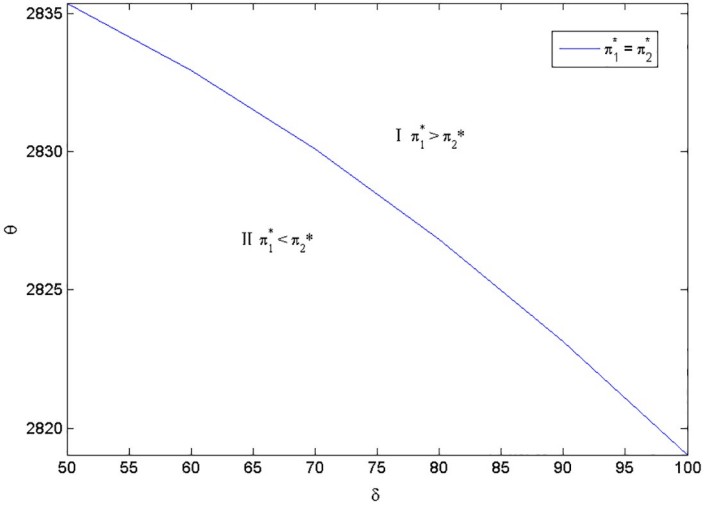

**Fig 3. Optimal naked vehicle common components selection under $c_1 = 1300$.**

## Batteries as common components

In this subsection, we consider that the manufacturer makes the battery as a commonality, and the common battery could be high quality or low quality.

**Scenario 3: High-quality batteries as common components.**    When the EV manufacturer provides two types of EVs to the market and the two products has the same high quality battery, then the demand functions of the product with high quality vehicle and battery and the product with low quality vehicle and high quality battery are as follows:

$$D_1 = a_1 - p_1 - \theta(1 - q_2^H) + \delta r \tag{17}$$

$$D_2 = a_2 - p_2 - \theta(1 - q_2^H) + \delta r \tag{18}$$

The manufacturer's profit is:

$$\pi_3 = (p_1 - C_1)D_1 + (p_2 - C_2)D_2 - e_2 - \frac{1}{2}kr^2 \tag{19}$$

where,

$$C_1 = c_1 q_1^{H^2} + c_2 q_2^{H^2} - c'\sqrt{e_2} \;\; = \;\; c^{HH} - c'\sqrt{e_2} \tag{20}$$

$$C_2 = c_1 q_1^{L^2} + c_2 q_2^{H^2} - c'\sqrt{e_2} \;\; = \;\; c^{LH} - c'\sqrt{e_2} \tag{21}$$

**Theorem 3.** In scenario 3, the optimal number of the charging stations, the prices are given as follows:

$$\ddot{r} = \frac{\delta(a_1 + a_2 - c^{HH} - c^{LH} - 2\theta + 2q_2^H\theta + 2c'\sqrt{e_2})}{2(k - \delta^2)} \tag{22}$$

$$\ddot{p}_1 = \frac{2k(a_1 + c^{HH} - \theta + q_2^H\theta - c'\sqrt{e_2}) + \delta^2(a_2 - a_1 - 3c^{HH} - c^{LH} + 4c'\sqrt{e_2})}{4(k - \delta^2)} \tag{23}$$

$$\ddot{p}_2 = \frac{2k(a_2 + c^{LH} - \theta + q_2^H\theta - c'\sqrt{e_2}) + \delta^2(a_1 - a_2 - c^{HH} - 3c^{LH} + 4c'\sqrt{e_2})}{4(k - \delta^2)} \tag{24}$$

**Scenario 4: Low-quality batteries as common components.** When the EV manufacturer provides two types of EVs to the market and the two products has the same low quality battery, then the demand functions are as follows:

$$D_1 = a_1 - p_1 - \theta(1 - q_2^L) + \delta r \tag{25}$$

$$D_2 = a_2 - p_2 - \theta(1 - q_2^L) + \delta r \tag{26}$$

The manufacturer's profit is:

$$\pi_4 = (p_1 - C_1)D_1 + (p_2 - C_2)D_2 - e_2 - \frac{1}{2}kr^2 \tag{27}$$

where,

$$C_1 = c_1 q_1^{H^2} + c_2 q_2^{L^2} - c'\sqrt{e_2} = c^{HL} - c'\sqrt{e_2} \tag{28}$$

$$C_2 = c_1 q_1^{L^2} + c_2 q_2^{L^2} - c'\sqrt{e_2} = c^{LL} - c'\sqrt{e_2} \tag{29}$$

**Theorem 4.** In scenario 4, the optimal number of charging stations, the EVs' prices are given as follows:

$$\ddot{r} = \frac{\delta(a_1 + a_2 - c^{HL} - c^{LL} - 2\theta + 2q_2^L\theta + 2c'\sqrt{e_2})}{2(k - \delta^2)} \tag{30}$$

$$\ddot{p}_1 = \frac{2k(a_1 + c^{HL} - \theta + q_2^L\theta - c'\sqrt{e_2}) + \delta^2(a_2 - a_1 - 3c^{HL} - c^{LL} + 4c'\sqrt{e_2})}{4(k - \delta^2)} \tag{31}$$

$$\ddot{p}_2 = \frac{2k(a_2 + c^{LL} - \theta + q_2^L\theta - c'\sqrt{e_2}) + \delta^2(a_1 - a_2 - c^{HL} - 3c^{LL} + 4c'\sqrt{e_2})}{4(k - \delta^2)} \tag{32}$$

By comparing the equilibrium solutions, we can obtain the following propositions.

**Proposition 6.** In scenarios 3 and 4,

(1) if $\theta > c_2(q_2^H + q_2^L)$, then $\ddot{r} > \ddot{r}$; if $\theta < c_2(q_2^H + q_2^L)$, then $\ddot{r} < \ddot{r}$;

(2) if $0 < \delta < \sqrt{\frac{k}{2}}$, then $\ddot{p}_1 > \ddot{p}_1, \ddot{p}_2 > \ddot{p}_2$; if $\sqrt{\frac{k}{2}} < \delta < \sqrt{k}$, then

a) if $\theta > \frac{c_2(2\delta^2 - k)(q_2^H + q_2^L)}{k}$, then $\ddot{p}_1 > \dddot{p}_1, \ddot{p}_2 > \dddot{p}_2$;

b) if $\theta < \frac{c_2(2\delta^2 - k)(q_2^H + q_2^L)}{k}$, then $\ddot{p}_1 < \dddot{p}_1, \ddot{p}_2 < \dddot{p}_2$;

(3) when $\theta > c_2(q_2^H + q_2^L)$, then $\ddot{D}_1 > \dddot{D}_1, \ddot{D}_2 > \dddot{D}_2$; when $\theta < c_2(q_2^H + q_2^L)$, then $\ddot{D}_1 < \dddot{D}_1$, $\ddot{D}_2 < \dddot{D}_2$.

Proposition 6 (1) shows that if the consumers' range anxiety towards battery is much bigger than a certain level ($\theta > c_2(q_2^H + q_2^L)$), then the EV manufacturer still need build more charging station even through the battery quality is high.

Proposition 6 (2) and (3) indicate that if consumers' range anxiety towards battery is above a threshold point ($\theta > c_2(q_2^H + q_2^L)$), then the demands and prices of products with high quality battery are larger than those with low quality battery. If the anxiety is in a middle level ($\frac{c_2(2\delta^2 - k)(q_2^H + q_2^L)}{k} < \theta < c_2(q_2^H + q_2^L)$), then the prices of the products with high quality battery are bigger than those with the low-quality battery, but the demands are smaller. When the anxiety towards battery is low ($\theta < \frac{c_2(2\delta^2 - k)(q_2^H + q_2^L)}{k}$), then the prices and demands of the products with low quality battery are bigger than those with high quality battery.

**Proposition 7.** In scenarios 3 and 4,

(1) $\ddot{r}$ and $\dddot{r}$ decrease with $\theta$;

(2) $\ddot{p}_1, \ddot{p}_2, \dddot{p}_1$ and $\dddot{p}_2$ decrease with $\theta$;

(3) $\ddot{D}_1, \ddot{D}_2, \dddot{D}_1$ and $\dddot{D}_2$ decrease with $\theta$.

**Proposition 8.** In scenarios 3 and 4,

(1) $\ddot{r}$ and $\dddot{r}$ increase with $c'$;

(2) when $\sqrt{\frac{k}{2}} < \delta < \sqrt{k}$, then $\ddot{p}_1, \ddot{p}_2, \dddot{p}_1$ and $\dddot{p}_2$ increase with; and when $0 < \delta < \sqrt{\frac{k}{2}}$, then $\ddot{p}_1$, $\ddot{p}_2, \dddot{p}_1$ and $\dddot{p}_2$ decrease with $c'$;

(3) $\ddot{D}_1, \ddot{D}_2, \dddot{D}_1$ and $\dddot{D}_2$ increase with $c'$.

Proposition 8(1) and (3) indicate that as unit product cost reduction due to the use of common parts increases, the optimal number of charging stations and demands increase. EVs' prices may increase or decrease, which depends on the relationship between the cost coefficient of the building charging station and the demand sensitivity towards the number of the charging station (Proposition 8(2)). If $\delta \geq \sqrt{k/2}$, then the prices increase; otherwise, the prices decrease.

**Proposition 9.** In scenarios 3 and 4,

(1) $\ddot{r}$ and $\dddot{r}$ increase with $\delta$;

(2) $\ddot{p}_1, \ddot{p}_2, \dddot{p}_1$ and $\dddot{p}_2$ increase with $\delta$;

(3) $\ddot{D}_1, \ddot{D}_2, \dddot{D}_1$ and $\dddot{D}_2$ increase with $\delta$.

**Proposition 10.** In scenarios 3 and 4,

(1) $\ddot{r}$ and $\dddot{r}$ decrease with $k$;

(2) $\ddot{p}_1, \ddot{p}_2, \dddot{p}_1$ and $\dddot{p}_2$ decrease with $k$;

(3) $\ddot{D}_1, \ddot{D}_2, \dddot{D}_1$ and $\dddot{D}_2$ decrease with $k$.

**Table 3. The trend of optimal decision changes with related parameters.**

| | | | $r$ | $p_1$ | $p_2$ |
|---|---|---|---|---|---|
| Scenarios 1 and 2 | $\theta$ | | ↓ | ↓ | ↓ |
| | $c$ | $0 < \delta < \sqrt{\frac{k}{2}}$ | ↑ | ↓ | ↓ |
| | | $\sqrt{\frac{k}{2}} < \delta < \sqrt{k}$ | ↑ | ↑ | ↑ |
| | $\delta$ | | ↑ | ↑ | ↑ |
| | $k$ | | ↓ | ↓ | ↓ |
| Scenarios 3 and 4 | $\theta$ | | ↓ | ↓ | ↓ |
| | $c'$ | $0 < \delta < \sqrt{\frac{k}{2}}$ | ↑ | ↓ | ↓ |
| | | $\sqrt{\frac{k}{2}} < \delta < \sqrt{k}$ | ↑ | ↑ | ↑ |
| | $\delta$ | | ↑ | ↑ | ↑ |
| | $k$ | | ↓ | ↓ | ↓ |

Same to Propositions 4 and 5, Propositions 9 and 10 show that the optimal solutions' change trends with the demand sensitivity towards the number of the charging station and the cost coefficient of the building charging station.

For convenience, Table 3 summarizes the change trends of optimal decisions with related parameters.

Considering the complexity of manufacturer's profit functions in Scenarios 3 and 4, similar to Figs 2 and 3, we use the numerical examples (Figs 4 and 5) to compare the profits in the two scenarios. The parameters are specified as follows: $a_1 = a_2 = 2000$, $e_2 = 1000$, $c_1 = 1000$, $q_1^H = 0.8$, $q_1^L = 0.5$, $q_2^H = 0.7$, $q_2^L = 0.5$, $c' = 20$, $k = 20000$.

Similar to Figs 2–5, show that the manufacturer's optimal product strategy with different consumers' anxieties towards battery and charging stations. When consumers' anxiety towards battery is very low, then the manufacturer could produce two types of EVs with low-quality battery as a common component (Zone I). otherwise, the manufacturer will make more money from producing EVs with high-quality battery (Zone II). The finding seems straightforward. Different from Figs 2 and 3, the product strategy in Figs 4 and 5 is only related to consumers' anxiety towards battery and not related to consumers' sensitivity towards charging stations.

## Discussions

In this subsection, we will compare the four scenarios and discuss in which scenario the manufacturer could make more money. Considering the complexity of the profit functions, we use the numerical examples to give the comparison.

The different common components for the manufacturer implies different unit product cost reduction and production line R&D investment. Hence, we discuss the effects of the cost saving of different common parts on the optimal selection of the manufacturer's production line. Let $a_1 = a_2 = 2000$, $e_1 = 800$, $e_2 = 1000$, $c_1 = 1000$, $c_2 = 2000$, $k = 20000$, $\delta = 20$. In Fig 6, $q_1^H = 0.8$, $q_1^L = 0.5$, $q_2^L = 0.5$, $\theta = 1200$. In Figs 7 and 8, $q_1^H = 0.5$, $q_1^L = 0.2$, $q_2^H = 0.6$, $q_2^L = 0.3$.

Fig 6 shows that when the cost reduction coefficient of the battery as the common component is relatively big, using battery as the common component can make the manufacturer get more profit (Zone I and II). When the cost reduction coefficient of using battery as the

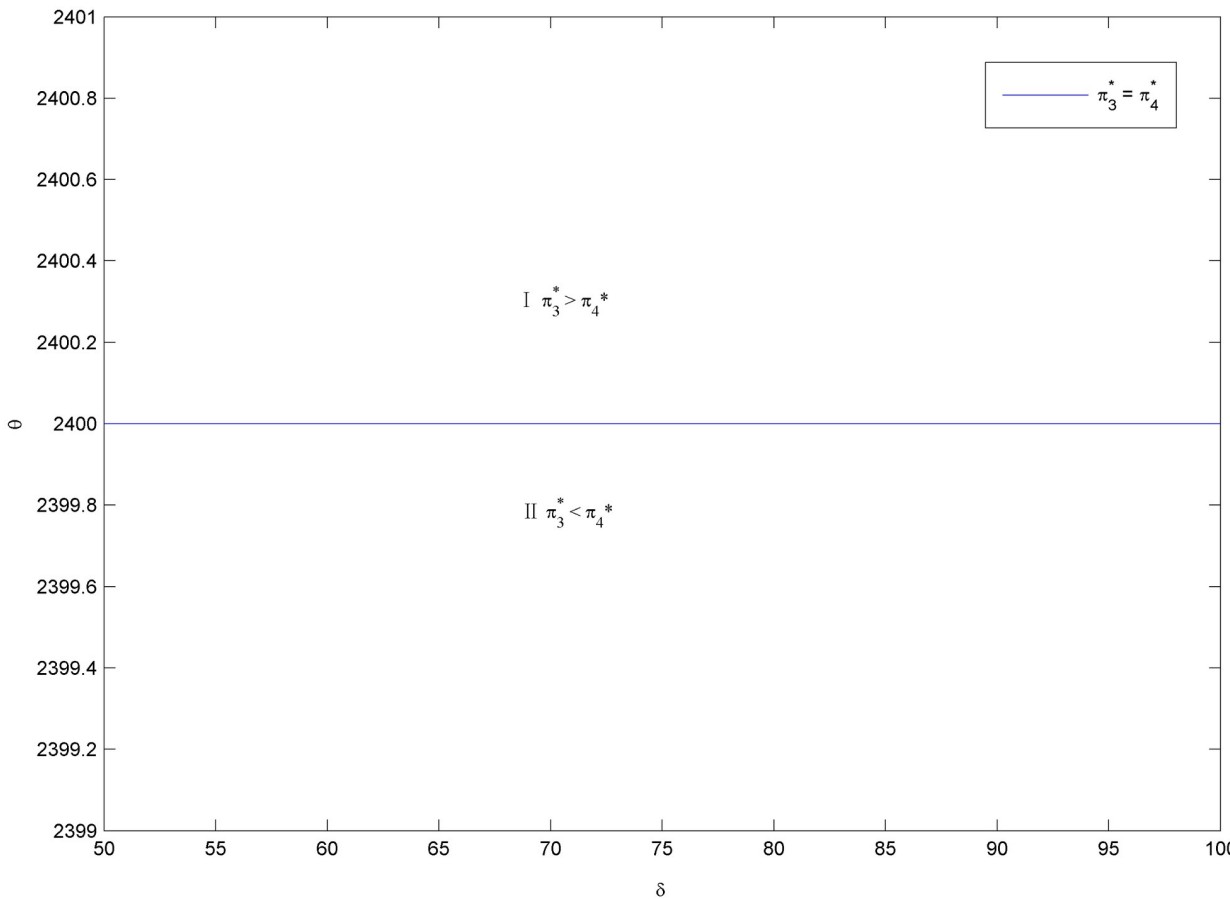

**Fig 4. Optimal battery common components selection under $c_2 = 2000$.**

common component is very small, using naked vehicle as the common component may make the manufacturer get more profit (Zone III, IV and V). And the cost saving coefficient of battery common parts has greater influence on the selection of commonality. In general, low quality commonality could make manufacturers gain more profits. When the cost reduction coefficient of using battery as the common component and using naked vehicle as the common component are at two extremes, the manufacturer then should choose suitable components as common components rather than considering the quality selection of components.

Figs 7 and 8 present the impact of consumer range anxiety towards battery on manufacturers' optimal profit. When consumers' range anxiety level is very low, choosing low-quality naked vehicles or battery as common components could get more profit. However, when consumers' range anxiety towards battery is very high, manufacturers should choose low-quality naked vehicles or high-quality battery as common components to maximize profits. This is just in line with the reality that the sales volume of Wuling Hongguang mini EVs with low naked vehicle quality and Tesla with high battery quality rank among the top three in the world. In sum, it is more likely to obtain high profits by using low-quality naked vehicle as commonality.

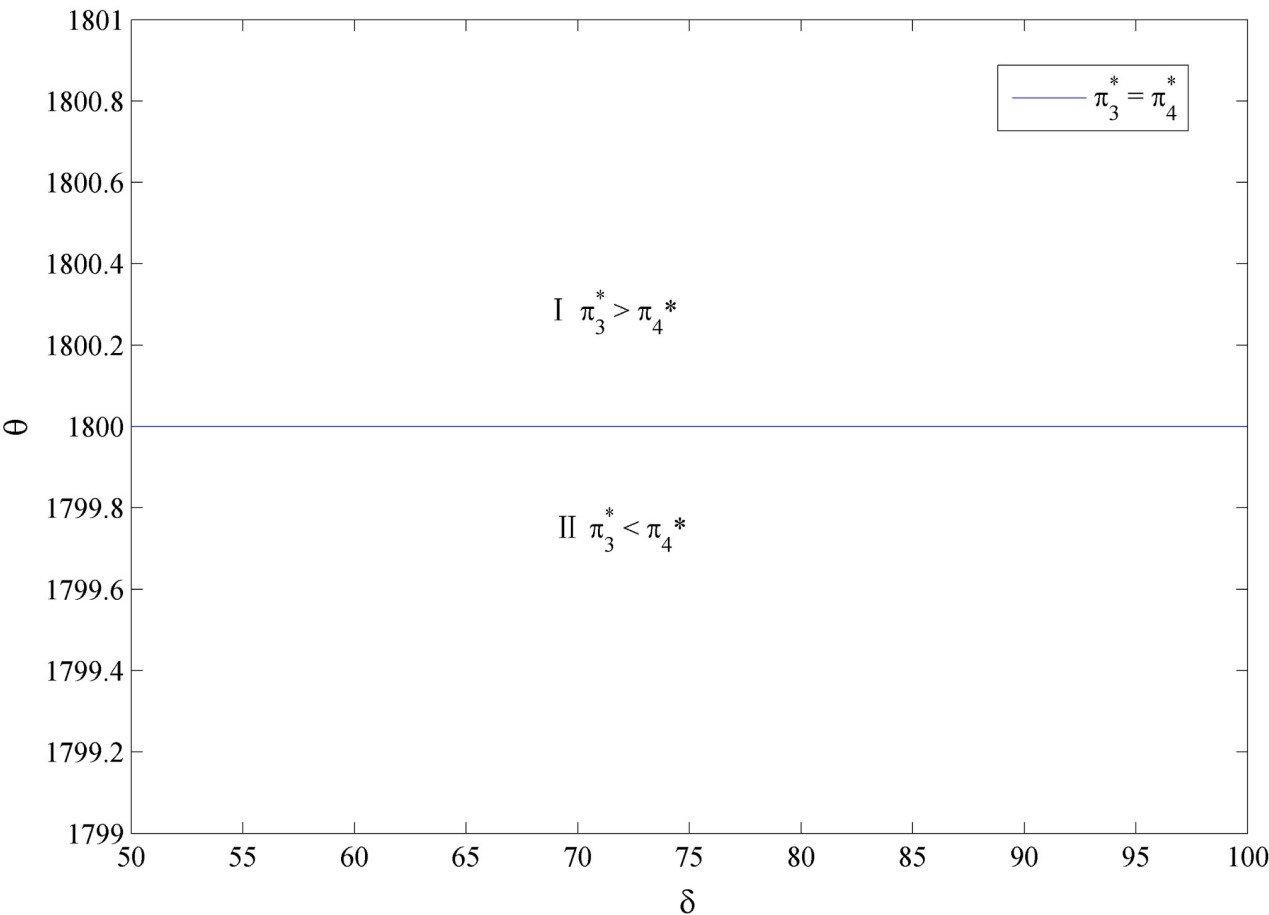

**Fig 5. Optimal battery common components selection under $c_2 = 1500$.**

## Conclusion

Assume one manufacturer provides two types of EVs, in this paper we divide electric vehicles into two parts–naked vehicle and battery, to explore the component commonality choice, and the naked vehicle and battery quality configuration. In addition, considering consumers' sensitivity towards the charging convenience, we also discuss the optimal number of the charging station built by the EV manufacturer.

## Managerial insights

We obtain some interesting managerial insights: first, high consumers' range anxiety towards battery will induce high demands and prices of EV with high quality battery. Hence, facing high consumer range anxiety, manufacturers should produce EV with high-quality battery. As the anxiety decreases, then the demands of the EV with high quality battery lower. When the anxiety towards battery is very low, then the prices and demands of the product with low quality battery are even bigger than those with high quality battery as the number of the charging stations increases. Therefore, when the EV manufacturer determines the EVs' battery configuration, identifying consumer range anxiety towards battery in the target market is very significant.

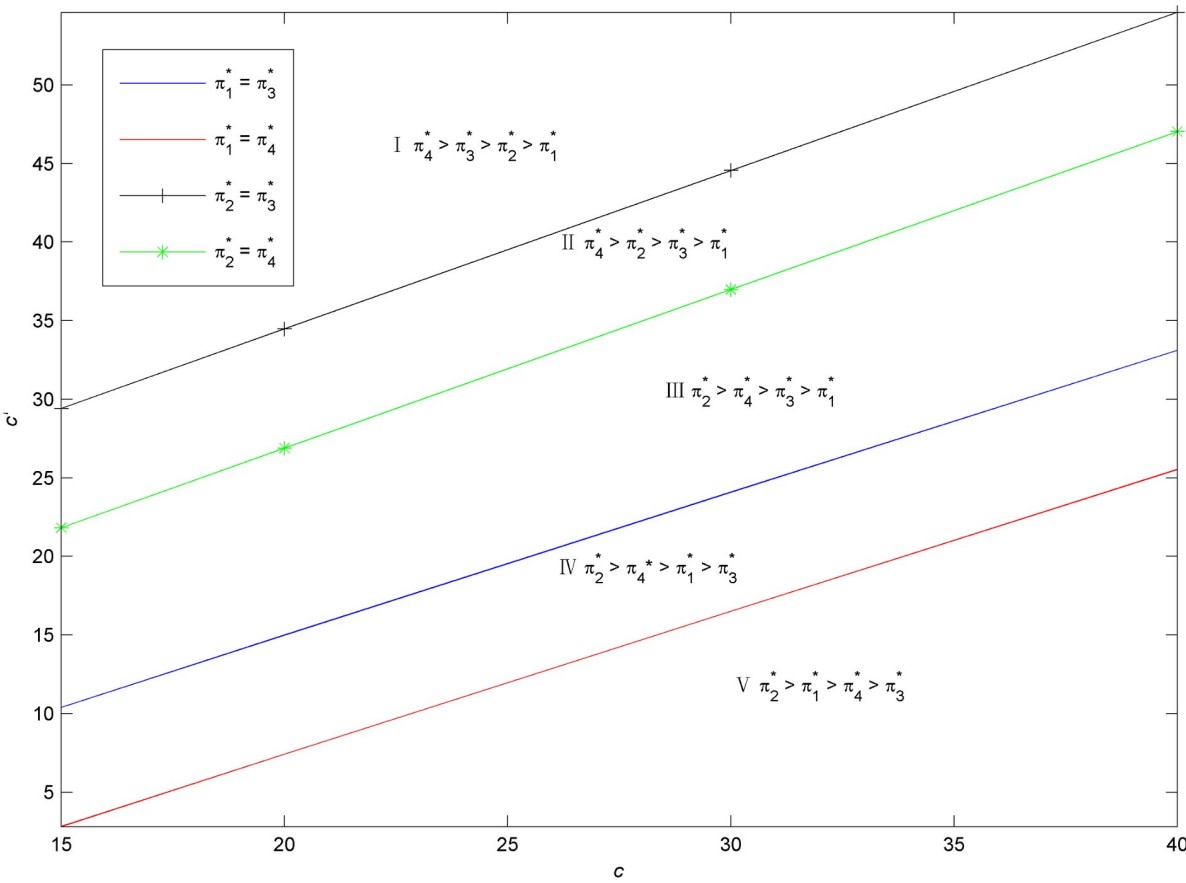

**Fig 6. Partion of the $c - c'$. as $q_{I_2}^H = 0.7$.**

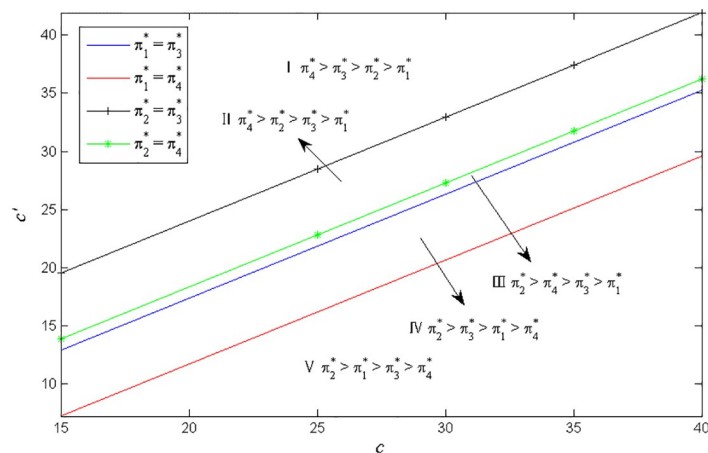

**Fig 7. Partion of the $c - c'$ as $\theta = 1200$.**

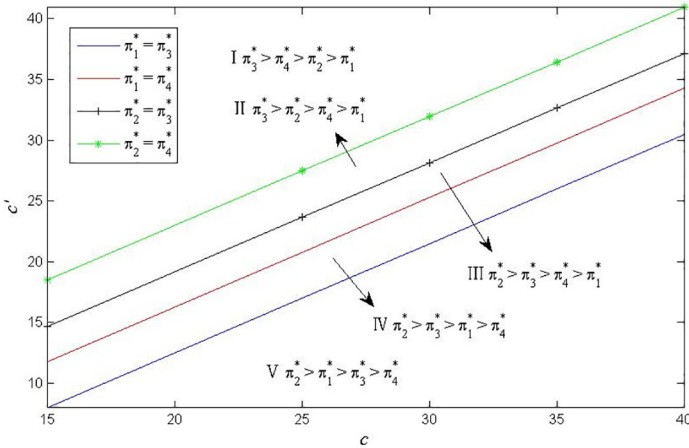

**Fig 8. Partion of the $c - c'$ as $\theta = 2200$.**

Second, the larger consumers' sensitivity towards charging station and range anxiety towards battery, the higher EV configuration and prices. When consumers are very concerned about the battery range and charging convenience, the manufacturer needs provide the high-end electric vehicles using the high quality naked vehicle as commonality, such as Tesla. If consumers don't have much anxiety about mileage, for example, they purchase EV as a short distance courtesy car, then the manufacturer could configure the EVs using low quality naked vehicle as the common part, such as Hongguang mini EV. Hence, before the EV manufacturer enters to the market, it is very important to determine the product position and configuration.

Third, the low quality naked vehicle as common component will increase both the number of the charging stations and the demand. This is a good way to increase the number of charging stations, that is, increasing the sales of low-end electric vehicles to drive the construction of charging stations. Hence, the government could encourage the manufacturer to develop low-end EVs that have huge potential market demand to stimulate the charging station's construction.

Fourth, if one manufacturer who produces EVs with low quality naked vehicle and builds the charging station may ask for much higher EV price than the manufacturer who provides the high-end EV and does not construct charging station, which depends on the demand and cost increment incurred by increasing one charging station. For example, when the increased demand is relatively large and the cost is not very big incurred by one more charging station, then the low-end EVs producing by one manufacturer will have a higher price than the high-end EVs providing by the other manufacturer.

Finally, using the component commonality will increase both EVs demands and the number of the charging stations. But the EVs prices may not decrease as the number of the charging stations increases, which depends on the relationship between the demand increment incurred by one more charging station and the cost coefficient of building the charging station. In addition, there is no absolute profit advantage in making battery as common parts or naked vehicle as common parts. The choice of the common part is determined by the production cost coefficient saved by the common part, the R&D cost of the common part, the production cost of batteries and naked vehicle, and the consumers' mileage anxiety and so on. When consumers' range anxiety towards battery is very high, manufacturers should choose low-quality naked vehicles or high-quality battery as common components. The low quality naked vehicle

as common component will increase both the number of the charging stations and the demand and the manufacturer is more likely to obtain high profits.

## Limitations

In this paper, we only discuss the number of the charging stations built by the manufacturer, but do not consider the charging station built by the government or the third party, hence future study could investigate how the public charging infrastructure affects manufacturer's decisions. In addition, the number of the charging station and the EV demand interact with each other, but in this paper, we only incorporate the effects of the number of charging station on demand and do not consider how the demand increment affects the number of the number of the charging station. Above two limitations point out the two future research directions.

## Supporting information

**S1 Appendix.**
(DOCX)

## Author Contributions

**Conceptualization:** Wenchao Yu, Linghong Zhang.

**Formal analysis:** Wenchao Yu.

**Methodology:** Wenchao Yu.

**Supervision:** Linghong Zhang, Rui Lu.

**Writing – original draft:** Wenchao Yu.

**Writing – review & editing:** Linghong Zhang, Junjie Ma.

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
