## [Decision Letter · Decision Letter 0]

22 Dec 2022

PONE-D-22-26392Optimal number of charging station and pricing strategy for the electric vehicle with component commonality considering consumer range anxietyPLOS ONE

Dear Dr. Zhang,

Thank you for submitting your manuscript to PLOS ONE. After careful consideration, we feel that it has merit but does not fully meet PLOS ONE’s publication criteria as it currently stands. Therefore, we invite you to submit a revised version of the manuscript that addresses the points raised during the review process.

We look forward to receiving your revised manuscript.

Kind regards,

Sathishkumar V E

Academic Editor

PLOS ONE

Journal Requirements:

2. "PLOS requires an ORCID iD for the corresponding author in Editorial Manager on papers submitted after December 6th, 2016. Please ensure that you have an ORCID iD and that it is validated in Editorial Manager. To do this, go to ‘Update my Information’ (in the upper left-hand corner of the main menu), and click on the Fetch/Validate link next to the ORCID field. This will take you to the ORCID site and allow you to create a new iD or authenticate a pre-existing iD in Editorial Manager. Please see the following video for instructions on linking an ORCID iD to your Editorial Manager account: " ext-link-type="uri" xlink:type="simple">https://www.youtube.com/watch?v=_xcclfuvtxQ"

3. PLOS ONE does not copy edit accepted manuscripts (https://journals.plos.org/plosone/s/criteria-for-publication#loc-5). To that effect, please ensure that your submission is free of typos and grammatical errors

4. Your abstract cannot contain citations. Please only include citations in the body text of the manuscript, and ensure that they remain in ascending numerical order on first mention.

Reviewers' comments:

Reviewer's Responses to Questions

**Comments to the Author**

1. Is the manuscript technically sound, and do the data support the conclusions?

Reviewer #1: Partly

Reviewer #2: Yes

2. Has the statistical analysis been performed appropriately and rigorously? 

Reviewer #1: Yes

Reviewer #2: Yes

3. Have the authors made all data underlying the findings in their manuscript fully available?

Reviewer #1: No

Reviewer #2: Yes

4. Is the manuscript presented in an intelligible fashion and written in standard English?

Reviewer #1: Yes

Reviewer #2: Yes

5. Review Comments to the Author

Reviewer #1: 1.found 2 sub-sections with the same name "Electric vehicle" in the Literature Review, could you please give them a different name or merge the contents into one paragraph.

2. In a table, present the information derived from the formulas.

Reviewer #2: 1.Introduction section needs to be re-written to improve its quality and readability.

2.What is the motivation of the proposed work? Research gaps, objectives of the proposed work should be clearly justified

3.Overall, the basic background is not introduced well, where the notations are not illustrated much clear.

4.The literature has to be strongly updated with some relevant and recent papers focused on the fields dealt with in the manuscript.

5.The study lacks a theoretical framework which is important for the reader to grasp the crust of the research.

6. Explain why the current method was selected for the study, its importance and compare with traditional methods.

7.Authors are suggested to include more discussion on the results and also include some explanation regarding the justification to support why the proposed method is better in comparison towards other methods

8.Does this kind of study have never attempted before? Justify this statement and give an appropriate explanation to do so in this paper.

9. Improve the conclusion section

10. What are the limitations of the study?

6. PLOS authors have the option to publish the peer review history of their article (what does this mean?). If published, this will include your full peer review and any attached files.

Reviewer #1: No

Reviewer #2: **Yes: **Usha Moorthy

---

## [Author Response · Author response to Decision Letter 0]

28 Feb 2023

I uploaded a separate file named response to reviewers along with the manuscript.

---

## [Decision Letter · Decision Letter 1]

8 Mar 2023

Optimal number of charging station and pricing strategy for the electric vehicle with component commonality considering consumer range anxiety

PONE-D-22-26392R1

Dear Dr. Zhang,

We’re pleased to inform you that your manuscript has been judged scientifically suitable for publication and will be formally accepted for publication once it meets all outstanding technical requirements.

Kind regards,

Sathishkumar V E

Academic Editor

PLOS ONE

Additional Editor Comments (optional):

Reviewers' comments:

Reviewer's Responses to Questions

**Comments to the Author**

1. If the authors have adequately addressed your comments raised in a previous round of review and you feel that this manuscript is now acceptable for publication, you may indicate that here to bypass the “Comments to the Author” section, enter your conflict of interest statement in the “Confidential to Editor” section, and submit your "Accept" recommendation.

Reviewer #2: (No Response)

2. Is the manuscript technically sound, and do the data support the conclusions?

Reviewer #2: (No Response)

3. Has the statistical analysis been performed appropriately and rigorously? 

Reviewer #2: (No Response)

4. Have the authors made all data underlying the findings in their manuscript fully available?

Reviewer #2: (No Response)

5. Is the manuscript presented in an intelligible fashion and written in standard English?

Reviewer #2: (No Response)

6. Review Comments to the Author

Reviewer #2: (No Response)

7. PLOS authors have the option to publish the peer review history of their article (what does this mean?). If published, this will include your full peer review and any attached files.

Reviewer #2: **Yes: **Usha Moorthy

---

## [Editor Report · Acceptance letter]

24 Apr 2023

PONE-D-22-26392R1 

Optimal number of charging station and pricing strategy for the electric vehicle with component commonality considering consumer range anxiety 

Dear Dr. Zhang:

I'm pleased to inform you that your manuscript has been deemed suitable for publication in PLOS ONE. Congratulations! Your manuscript is now with our production department. 

Kind regards, 

on behalf of

Dr. Sathishkumar V E 

Academic Editor

PLOS ONE